# Slowdown of growth controls cellular differentiation

Jatin Narula[1,†], Anna Kuchina[2,†], Fang Zhang[2], Masaya Fujita[3], Gürol M Süel[2] & Oleg A Igoshin[1,*]

## Abstract

How can changes in growth rate affect the regulatory networks behavior and the outcomes of cellular differentiation? We address this question by focusing on starvation response in sporulating *Bacillus subtilis*. We show that the activity of sporulation master regulator Spo0A increases with decreasing cellular growth rate. Using a mathematical model of the phosphorelay—the network controlling Spo0A—we predict that this increase in Spo0A activity can be explained by the phosphorelay protein accumulation and lengthening of the period between chromosomal replication events caused by growth slowdown. As a result, only cells growing slower than a certain rate reach threshold Spo0A activity necessary for sporulation. This growth threshold model accurately predicts cell fates and explains the distribution of sporulation deferral times. We confirm our predictions experimentally and show that the concentration rather than activity of phosphorelay proteins is affected by the growth slowdown. We conclude that sensing the growth rates enables cells to indirectly detect starvation without the need for evaluating specific stress signals.

**Keywords** *B. subtilis*; phosphorelay; signal integration
**Subject Categories** Microbiology, Virology & Host Pathogen Interaction; Quantitative Biology & Dynamical Systems
**Mol Syst Biol. (2016) 12: 871**

## Introduction

Stress response networks in bacteria need to be able to sense environmental conditions to activate the appropriate response for their survival (Taylor & Zhulin, 1999; Storz *et al*, 2011). Many stress response networks sense and respond specifically to a particular environmental signal by employing a sensor protein that binds a specific signaling molecule and transduces this information to downstream network components (Laub & Goulian, 2007). However, some stress conditions, such as starvation, can be triggered by a wide range of environmental perturbations and therefore are hard to define in terms of the level of a single metabolite (Peterson *et al*, 2005). As a result, responding to starvation requires a gene regulatory network that can detect the availability of a variety of nutrients and integrate this information into the cellular response. Conversion of this type of multi-nutrient information into cellular responses is a complex task. Even for the best studied model systems, we do not fully understand how gene regulatory networks control bacterial cell survival in response to starvation.

*Bacillus subtilis* cells survive prolonged starvation by differentiating into stress-resistant and metabolically inert spores (Fig 1A; Higgins & Dworkin, 2012). This differentiation program, known as sporulation, is controlled by the master regulator Spo0A (0A) which is active in its phosphorylated (0A~P) form (Errington, 2003). 0A activity itself is regulated by a complex network known as the sporulation phosphorelay—a more complex version of the bacterial two-component regulatory systems (Burbulys *et al*, 1991). This phosphorelay (Fig 1B) consists of the major sporulation kinase KinA that autophosphorylates and indirectly transfers the phosphoryl group to 0A via the intermediate proteins Spo0F (0F) and Spo0B (0B).

The expression levels of *kinA*, *0F*, and *0A* are regulated by 0A~P via direct and indirect transcriptional feedback (Predich *et al*, 1992; Fujita & Sadaie, 1998). Crucially, autophosphorylation of KinA can be blocked by high levels of 0F (Fig 1B) through a substrate inhibition mechanism (Grimshaw *et al*, 1998; Chastanet *et al*, 2010; Narula *et al*, 2015). This substrate inhibition forms part of a negative feedback loop in the phosphorelay. Recently, we showed that this negative feedback allows cells to respond to transient gene dosage imbalance during DNA replication with a pulsatile activation of 0A (Narula *et al*, 2015). Due to the widely conserved arrangement of *0F* (326° -*oriC* proximal) and *kinA* (126°-*ter* proximal) genes on the chromosomes in *B. subtilis* and other sporulating bacteria, *0F* gene is replicated before that of *kinA*, leading to a transient decrease in the *kinA:0F* gene dosage ratio. Completion of DNA replication returns the ratio to 1:1 and triggers the phosphorelay to respond with a pulse of 0A~P. Thus, in every cell cycle of starving *B. subtilis* cells, completion of the DNA replication is followed by a pulse of 0A~P (Fig 1A). The decision to sporulate is based on the amplitude of the 0A~P

1 Department of Bioengineering, Rice University, Houston, TX, USA
2 Division of Biological Sciences, UCSD, San Diego, CA, USA
3 Department of Biology and Biochemistry, University of Houston, Houston, TX, USA
*Corresponding author. Tel: +1 713 348 5502; E-mail: igoshin@rice.edu
†These authors contributed equally to this work

pulse. Low-amplitude 0A~P pulses allow cells to divide medially and continue growth (Fig 1A—left), whereas when this amplitude exceeds a threshold (Fig 1A—right), cells divide asymmetrically and commit to sporulation (Fujita & Losick, 2005; Veening *et al*, 2009; Eswaramoorthy *et al*, 2010a; Narula *et al*, 2012).

Despite these developments in the understanding of the phosphorelay dynamics, the relationship between starvation and the amplitude of 0A~P pulses remains unclear. It has been suggested that a multi-protein phosphorelay, offering multiple entry points for putative starvation signals, is well suited for proper 0A activity regulation (Hoch, 1993; Ireton *et al*, 1993). However, it is unclear how the design of the phosphorelay enables it to combine information about several different essential nutrients (Dawes & Mandelstam, 1970), evaluate the starvation level, and correctly time the cell-fate decision to allow complete execution of a multistage sporulation program. As a result, the central question of how *B. subtilis* sporulation program senses nutrient levels remains open.

Here, we identify and explore the correlation between cell growth rates and amplitudes of 0A~P pulses. Using a combination of mathematical modeling and quantitative single-cell experiments, we uncover the mechanistic basis of this correlation. Further, we demonstrate that this relationship represents a strikingly simple way for the sporulation network to sense and integrate information about nutrient in order to decide between continuing vegetative growth and committing to sporulation.

## Results

### 0A~P pulse amplitudes are correlated with cell growth rate

To understand the dynamics of the *B. subtilis* starvation response, we used time-lapse microscopy to track single cells as they grow and sporulate in nutrient-limited media. In these conditions, *B. subtilis* cells do not sporulate immediately upon exposure to starvation. Instead, cells proceed with multiple rounds of vegetative division before eventually dividing asymmetrically and forming a spore. During this multi-cycle progression toward spore formation, cell growth rate (inferred from cell elongation rate) gradually decreases (Fig 1C).

To understand 0A activity dynamics in single cells during this period, we used fluorescent reporters to measure gene expression from 0A~P-regulated promoters for *0A* and *spo0F* ($P_{0A}$ and $P_{0F}$; see Materials and Methods). As expected from prior work, our measurements revealed that 0A activity (defined as the production rate of fluorescent reporter proteins) pulses during every cell cycle in starvation conditions (Fig 1E). $P_{0F}$ promoter activity similarly pulses once every cell cycle in starvation conditions (Fig EV1A–C). In contrast, measurements of the production rate of a fluorescent protein, YFP, expressed from an IPTG-inducible promoter ($P_{hsp}$) under different inducer concentration in starving cells showed no pulses (Fig EV1D–F). Our measurements further show that 0A activity pulse amplitudes increase gradually over multiple cell cycles. Notably, this increase in amplitudes coincides with the decrease in cell growth rate (compare Fig 1C and E). Quantification of this relationship using a Spearman's rank correlation showed a highly significant anti-correlation between 0A activity pulse amplitudes and cell growth rate ($\rho = -0.8$, *P*-value = 4e-64, *N* = 307; Fig 1F).

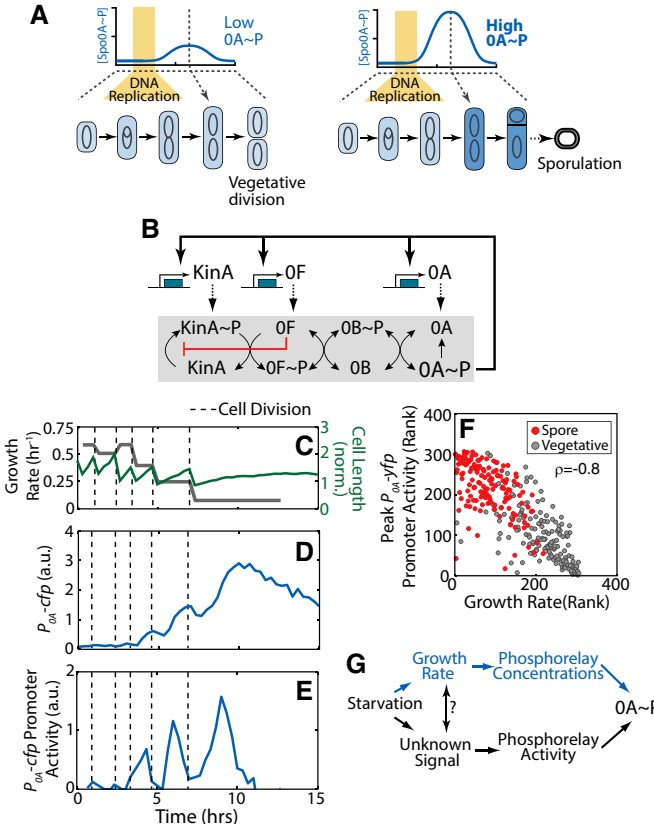

**Figure 1.  Decision making during *Bacillus subtilis* sporulation.**

A Sporulation commitment depends on the amplitude of a cell-cycle-coordinated pulse of the sporulation master regulator SpoO A~P (0A~P). Yellow bars indicate the DNA replication phase.

B The sporulation phosphorelay network that controls 0A~P formation (see text for details).

C–E Single-cell time-lapse microscopy using a $P_{0A}$-*cfp* reporter for 0A~P. (C) Cell length (green) and its cell growth rate, that is, cell-cycle-averaged log-derivative, (gray), for a single cell traced over multiple cell cycles in starvation media. Expression level of $P_{0A}$-*cfp* (D) increases in non-monotonic fashion. Its promoter activity (defined as production rate, an indicator of 0A~P level) shows pulses with an increased amplitudes that is coordinated with a decrease in growth rate (E). In (C–E), vertical dashed lines indicate cell divisions.

F Measurements of $P_{0A}$-*yfp* promoter activity show that 0A~P pulse amplitudes and growth rates are anti-correlated. Each dot corresponds to ranked measurements of the $P_{0A}$-*yfp* promoter activity pulse amplitude and growth rate of an individual cell cycle. Red and gray dots indicate cell cycles that end in sporulation and vegetative division, respectively. The resulting Spearman's rank correlation $\rho = -0.8$, *P*-value < $10^{-60}$, *N* = 307.

G Two hypothetical mechanisms behind the observed correlation. Starvation may be detected by the sporulation network via growth rate modulation of phosphorelay concentrations or by modulation of phosphorelay activity by growth-rate-correlated signal/metabolite.

Source data are available online for this figure.

Moreover, as expected the cells that end up as spores (red dots on Fig 1F) have the highest 0A~P peaks and correspondingly slowest growth rates.

What can explain this correlation? Given that the amplitude of the Spo0A~P peaks is determined by the phosphorelay, one can

propose two (non-mutually exclusive) mechanisms (Fig 1G). In the first one, slowdown of growth directly affects the concentrations of phosphorelay proteins through growth-dependent changes in DNA replication, dilution, cell volume, and transcription/translation rates (Bipatnath *et al*, 1998; Klumpp *et al*, 2009). As a result, slowdown of growth could increase concentrations of phosphorelay proteins and lead to higher 0A activity. This type of indirect sensing is particularly appealing since it bypasses the need for any dedicated metabolite sensing. In the second mechanism, growth slowdown is correlated with accumulation of a certain intracellular signaling molecule ("unknown starvation signal") which activates one of the phosphorelay components and leads to Spo0A~P accumulation. For example, it has been suggested that the autophosphorylation activity of KinA may be modulated by the binding of ATP and NAD$^+$ to its PAS-A domain (Stephenson & Hoch, 2001; Kolodkin-Gal *et al*, 2013) although other studies indicate that the PAS domain is dispensable for the autophosphorylation activity (Eswaramoorthy *et al*, 2009; Winnen *et al*, 2013). The correlation of starvation signal with growth implies that the starvation signal regulates cell growth or that cell growth controls accumulation of the starvation signal or that both starvation signal and growth rate are controlled by a common upstream signal. In the following subsections, we evaluate the importance of these hypothetical mechanisms using a combination of mathematical modeling and single-cell experiments.

### Decrease in growth rate leads to accumulation of stable proteins

As cell volume increases exponentially, growth rate acts as an effective first-order degradation constant of all cellular proteins (Appendix Text A1). As a result, a decrease in the rate of growth leads to accumulation of constitutively produced proteins. For example, if a stable protein is produced at a constant rate of 10 molecules per min in a cell with 60 min generation time, its steady-state concentration would be 600 molecules/cell. If cell growth decreased to a generation time of 300 min, the concentration would increase to 3,000 molecules/cell. If on top of this, cell volume decreases for slower-growing cells, as is the case for starving *B. subtilis* cells, then that would further increase the concentration of the corresponding proteins. Combining these two effects into a simple mathematical model, we would predict the following dependence of concentration *C* on growth rate μ (see Appendix Text A1 for derivation):

$$C(\mu) = \frac{P}{V(\mu)(k_{\mathrm{deg}} + \mu)} \tag{1}$$

Here, $V(\mu)$ is a growth-rate-dependent cell volume, $k_{\mathrm{deg}}$ is the rate of protein degradation or deactivation and $P$ is a rate of protein production which for simplicity is assumed to be growth independent.

To test this model experimentally, we measured the single-cell level of a fluorescent reporter protein of cells growing with different rates and compared the results to the predictions of our equation (1). Phosphorelay proteins have been shown to accumulate as cells slow down their growth in starvation (Veening *et al*, 2008; Eswaramoorthy *et al*, 2010b; Levine *et al*, 2012), but feedback loops from 0A~P do not allow us to separate the direct effects of growth rate slowdown from the effects of increased 0A~P. Therefore, we

chose to test equation (1) by measuring the growth rate-dependent change in the level of a stable fluorescent protein, YFP, expressed from an IPTG-inducible promoter ($P_{hsp}$) under different inducer concentrations in starving *B. subtilis*. The results demonstrate that a decrease in growth rate during starvation can increase the level of stable proteins severalfold and that relative increase is independent of the rate of production as data collapses upon normalization (Fig 2A). When these results are compared to the model, we can see that the trend can be fitted to equation (1) leading to the value of $k_{\mathrm{deg}} = 0.12$/h (about 6-h half-life).

### Accumulation of phosphorelay proteins with growth slowdown is sufficient to explain observed increase in 0A~P levels

To understand how growth rate affects the amplitude of 0A~P pulses, we employed a detailed mathematical model of the sporulation phosphorelay that we have recently developed (Narula *et al*, 2015). This model successfully explained both the mechanism and timing of 0A~P pulsing but postulated an increase in KinA autophosphorylation rate to explain the increase in pulse amplitudes during starvation. Here, we instead chose to keep all the biochemical rate constants (including KinA autophosphorylation rate) fixed and instead study the effects of reduction of the dilution rate and cell volume on 0A~P pulsing.

Our model simulations showed that these effects lead to increased 0A~P amplitudes as growth slows down and cell cycles get longer (Fig 2B). The results indicate that growth affects 0A~P pulsing in two major ways. At high growth rates, the DNA replication period takes all or most of the cell-cycle time and leaves little time for the *kinA:0F* gene dosage to be at a 1:1 ratio which in our model corresponds to the time of 0A~P pulsatile increase. As a result of subsequent round of DNA replication, gene *kinA:0F* dosage ratio returns to a 1:2 ratio and the pulses are cut short before reaching their maximal amplitude. At lower growth rates, each pulse reaches its peak amplitude before being brought down by the negative feedback in the network. Moreover, the decreases in cell volume and dilution rate at these lower growth rates lead to an increase in phosphorelay protein concentrations and consequently higher 0A~P pulse amplitudes (Fig 2B).

To determine how the increase in phosphorelay protein levels affects 0A~P and identify the phosphorelay proteins that play the most significant role in this increase, we performed a sensitivity analysis of our model. To this end, we chose to computationally change the production rates of individual phosphorelay proteins and examine how this affects 0A~P pulse amplitudes (Fig 2C). Our model indicated that pulse amplitudes are most sensitive to KinA (~25% change in peak 0A~P with 10% change in KinA). Increasing 0F also has a strong effect; however, in contrast to KinA, 0F decreases pulse amplitudes (increase in 0F by 10% decreases 0A~P peak by ~18%). Thus, our mathematical model predicts that a slowdown of cell growth over multiple cell cycles leads to a gradual increase in 0A~P peak levels (Fig 2D, solid line) mainly due to an increase in KinA concentration with decrease in cell volume and dilution.

Comparison of the model predictions (Fig 2D, solid line) with experimental data from time-lapse microscopy (Fig 2D, dots) showed that the effects of growth slowdown are sufficient to explain observed increases in peak amplitudes. The ultrasensitive dependence of 0A~P pulse amplitudes on growth rate is consistent with the trends in the data. Moreover, examining the distribution of

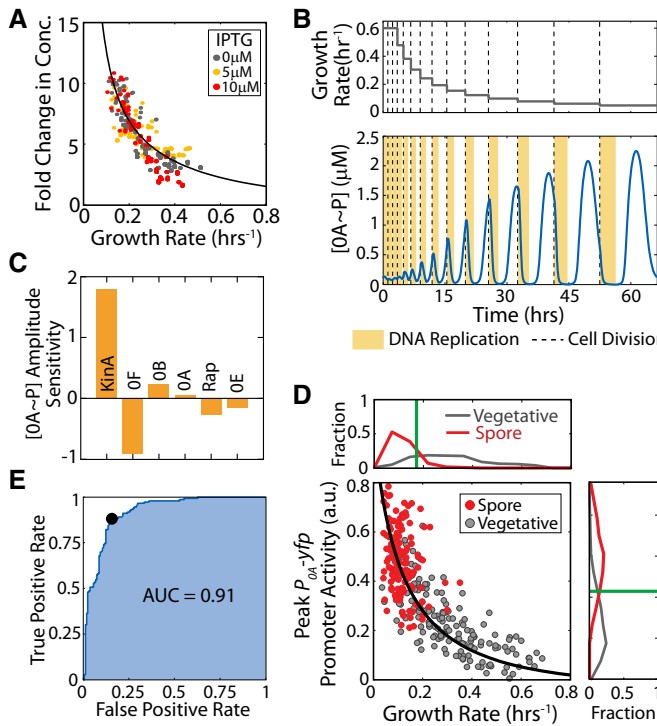

**Figure 2. Decrease in growth rate affects 0A~P levels and controls cell fate.**

A   Fold change in $P_{hsp}$-*yfp* fluorescence levels as a function of growth rate for different IPTG inducer concentrations. Black line represents a model fit for the effect of growth rate on the concentration of stable proteins.

B   Model time course of 0A~P during starvation showing cell-cycle-coordinated pulsing. Upper and lower panels show the growth rate (input) and 0A~P response (output), respectively. Yellow bars and dashed lines represent DNA replication periods and cell division, respectively. The effect of growth rate on protein levels was modeled following the results shown in (A). The simulation predicts that 0A~P pulse amplitudes increase with decreasing cell growth rate.

C   Sensitivity of the growth rate—0A~P pulse amplitude relationship to changes in the gene expression of phosphorelay components. Model simulations showed that 0A~P pulse amplitudes were most sensitive to changes in the production rate of kinase KinA. Increase in 0F caused a significant decrease in 0A~P pulse amplitude. 0A~P was found to be relatively insensitive to changes in other phosphorelay components.

D   Measurements of $P_{OA}$-*yfp* promoter activity confirm that 0A activity pulse amplitudes increase as growth rate decreases during starvation. Each dot corresponds to a single cell cycle. Gray and red dots correspond to cell cycles that end in vegetative division and spores, respectively. Solid lines show the model predictions. Panels to the right and bottom show histograms of $P_{OA}$-*yfp* promoter activity and (growth rate)$^{-1}$ for cell cycles that end in vegetative division (gray) and spores (red) for each strain. Vertical green lines show the thresholds that can be used to predict cell fate in each case.

E   Receiver operating characteristic (ROC) curve for growth-based cell-fate prediction. Blue line shows the relation between false-positive rate and true-positive rate for different values of growth rate threshold. The black dot marks the optimal growth rate threshold that minimizes classification error. The high area under the ROC curve (AUC = 0.91) indicates that growth rate is a highly robust predictor of sporulation.

Source data are available online for this figure.

0A~P peak activity and growth rate in the single-cell data (Fig 2D, histograms on the axes), we can see that values for sporulating and non-sporulating cells are well separated. This supports our

hypothesis that the slowdown of growth can act as a starvation signal and control the sporulation decision.

Since a 0A activity threshold is known to control sporulation cell fate (Chung *et al*, 1994; Fujita *et al*, 2005; Narula *et al*, 2012), we expect from the results in Fig 2D that this threshold would correspond to a threshold growth rate below which cells will sporulate. To test this, we constructed a cell-fate classifier by performing logistic regression to estimate the probability of sporulation as a function of peak $P_{OA}$ promoter activity or as a function of growth rate. Our results demonstrated an ultrasensitive dependence of the probability of sporulation on both metrics (Fig EV2). This observed ultrasensitive dependence allows us to define a threshold value for $P_{OA}$ promoter activity above which cells sporulate with at least 50% probability. Similarly, we can estimate a threshold growth rate below which cells sporulate. We can use these thresholds to predict whether a given cell cycle will end up producing a spore based on $P_{OA}$ promoter activity being above threshold (or, growth rate being below its respective the threshold). We found that the resulting predictions work very well, producing an accuracy of 85% for $P_{OA}$ promoter activity and 81% for growth rate.

Analysis of the relationship between growth rates of mother–daughter pairs and daughter cell fates showed that mother and daughter growth rates are weakly correlated ($\rho = 0.41$, $P$-value = 4.5e-14, $N_{pair} = 312$; Fig EV3A), suggesting that slow-growing mothers produce slow-growing daughter cells that are likely to sporulate. Furthermore, the correlation between sister cell growth rates is also quite weak ($\rho = 0.36$, $P$-value = 4e-06, $N_{pair} = 156$; Fig EV3B), indicating that there is a large chance that only one daughter cell will sporulate. The same growth threshold calculated in Fig 2D is able to accurately distinguish sister pairs with asymmetric fates and pairs with symmetric cell fates. Thus, the heterogeneity of growth during starvation has a significant impact on the cell fates adopted by sister cells, but the growth threshold model is still able to determine sporulation cell fate. Altogether, this analysis confirms that growth rate is indeed an accurate predictor of cell fate.

To test the robustness of growth rate as a cell-fate predictor for sporulation, we examined how the threshold value of this predictor affects its accuracy. We varied the growth rate threshold value (Fig 2E) to calculate the receiver operating characteristic (ROC) curve that provides the relation between false-positive rate (fraction of cells below growth rate threshold that remain vegetative) and true-positive rate (fraction of cells below growth rate threshold that sporulate). The overall goodness of a predictor is usually characterized by the area under the ROC curve (AUC; Fig 2E—shaded region). An AUC of 1 represents a perfect predictor, whereas AUC of 0.5 means the cell-fate prediction is no better than random (Hosmer & Lemeshow, 2000). In our case, the area is 0.91 indicating that growth rate is a highly robust predictor of sporulation. Thus, we conclude that a simple growth rate threshold (Fig 2D, green vertical line) is an accurate and robust predictor of sporulation cell fate.

**Growth slowdown dynamics control sporulation deferral**

In order to further test the effectiveness of growth rate as a predictor of cell fates, we investigated whether the multi-cell cycle deferral of sporulation can be explained simply by the dynamics of growth slowdown. To test this hypothesis, we tracked lineages of

sporulating cells (Fig 3A) to determine the fraction of cells that sporulated in each generation (Fig 3B, red dots). We found that the fraction of cells that decide to sporulate gradually increased from ~10% in the first generation to about ~65% in the fourth generation and remained approximately constant thereafter. To determine whether the growth rate threshold can explain this sporulation deferral, we quantified the distribution of cell growth rates for each generation of cells (Fig 3C) and calculated the fraction of cells with a growth rate below the sporulation threshold determined in Fig 2C. The resulting prediction is in good agreement with experimentally observed fractions (Fig 3B).

Next, we perturbed growth dynamics with nutrient addition to test the ability of the growth threshold model to explain cell-fate decisions in different conditions. Based on a simple population dynamics model, we expected that the addition of a relatively small dose of nutrients to the medium at the start of the experiment would delay the onset of starvation but lead to a faster decrease in cell growth rates once the increasing cell population had depleted the additional nutrients (see Appendix Fig S1, Appendix Text A2). In agreement with this model, experimental results (Fig 3D–F) showed that the addition of 0.0025% glucose indeed made the cells grow faster initially, followed by a rapid decrease in growth rate. Our model posits that the growth rate threshold is a function of biochemical parameters of the phosphorelay and not affected by nutrient addition. Accordingly, we used the same threshold as before to calculate fractions of cells that should sporulate in each generation. The results are in excellent agreement with the observations (Fig 3E), confirming that the dynamics of cell growth slowdown control the deferral of sporulation. Notably, the same threshold value robustly predicts fates of cells sporulating early and late in our experiment regardless of initial nutrient addition. This observation calls into question direct modulation of phosphorelay protein activity by metabolites and instead further reinforces the idea that cell growth rate is the primary signal that determines cell fate during starvation.

In light of these results, we reasoned that if cell growth rate controls the cell-fate decision, selective induction of 0A~P pulses under conditions of very low growth rates should lead to immediate and synchronized sporulation without any significant deferral. To test this idea, we used an engineered strain *iTrans-0F* (Fig 4A) in which we translocated the native copy of *0F* from its *oriC*-proximal location to the terminus and integrated an additional IPTG-inducible copy of *0F* close to the chromosome origin (Narula *et al*, 2015). In the wild-type strain, the *kinA:0F* gene dosage ratio decreases to 1:2 during DNA replication and then increases to 1:1 once replication is completed. These changes in *kinA:0F* gene dosage lead to a 0A~P decrease followed by an overshoot pulse of 0A activity. In *iTrans-0F* strain, the pulsing becomes IPTG dependent. Without IPTG, the inducible *oriC*-proximal copy of *0F* is inactive, *kinA:0F* gene dosage ratio remains 1:1 and DNA replication cannot trigger 0A~P pulses in this strain. As a result, this strain does not produce 0A~P pulses and does not sporulate without IPTG. Our modeling results predict that high levels of 0F induction in this strain result in 0F overexpression, which inhibits KinA autophosphorylation and 0A activation. In contrast, at low level, induction of *0F* from the *oriC*-proximal locus introduces a transient imbalance of gene dosage during DNA replication and thereby rescues 0A~P pulsing (Fig 4B). Moreover, we found that in *iTrans-0F* upon *0F* induction, 0A~P pulse amplitudes

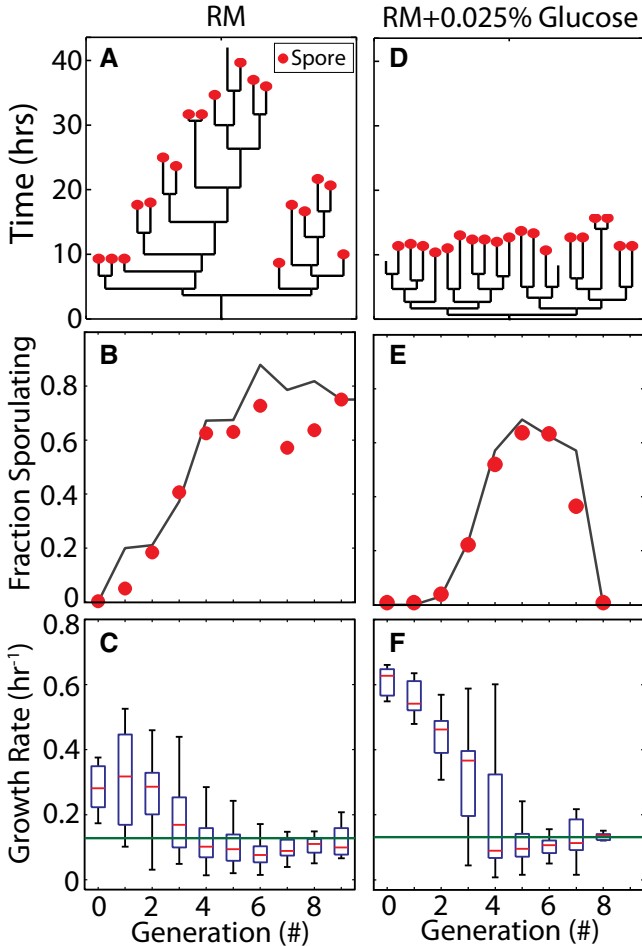

**Figure 3. Growth slowdown controls sporulation deferral.**

A     On normal sporulation media (resuspension media—RM) plates, sporulation timing (spores are marked by red dots) is heterogeneous and unsynchronized as demonstrated by a sample lineage.

B     Predictions of fraction of cells sporulating in each generation based on a growth rate threshold (black line) show good agreement with observed (red dots) fraction of sporulating cells for each generation. The growth rate threshold used was the same as threshold estimated from Fig 2D.

C     Distributions of growth rates in each cell cycle during starvation in RM. Box plots indicate median growth rate (red line), 25–75% quintile (blue box), and the range (black whiskers) of growth rates for each generation.

D–F   Same panels as (A–C) but with 0.0025% of glucose is added to the plates. (D) Sample lineage shows that addition of a small amount of glucose delays sporulation for several generations but decreases the heterogeneity in sporulation timing. (E) Growth threshold model with the same growth rate threshold as (B) shows excellent agreement between predicted (black line) and observed (red dots) fraction of sporulating cells. (F) Addition of the glucose increases the initial cell growth rate and delays the onset of starvation, but the resulting increase in cell number exhausts the nutrients leading to sharper decrease in growth rates.

Source data are available online for this figure.

increase gradually over multiple cell cycles as a function of cell growth rate similar to WT (compare Figs 2B and 4B). The model also showed that if induction of *0F* from the *oriC*-proximal locus is delayed until growth has slowed down, 0A~P amplitudes increase immediately without the multi-cell-cycle delay (Fig 4C). Thus, we

expect that cells should sporulate immediately upon delayed induction of *0F*.

We tested this prediction by varying the timing of IPTG addition for the *iTrans-0F* strain. As shown in Fig 4D and E, our measurements confirmed that 0A~P pulse amplitudes increase gradually if IPTG is added at the start of starvation, whereas they increase rapidly if IPTG is added at later times after the growth rate has fallen below the sporulation threshold. Further, sporulation was deferred and its deferral was heterogeneous when IPTG was added early (Fig 4F and I). In contrast, as predicted, when IPTG was added late, cells sporulated immediately upon IPTG addition without any deferral in a highly synchronized fashion (Fig 4G and I). Crucially, the timing of IPTG addition affected the 0A activity dynamics but not the average growth dynamics (Fig 4H), which rules out the possibility that 0A~P controls changes in growth rather than growth controlling 0A~P.

Altogether, these results provide further support for the hypothesis that slowdown of growth during starvation is the primary signal for sporulation and that it controls cell fate by modulating the amplitude of 0A~P pulse amplitudes.

### KinA activity does not depend on the growth rate

Our results thus far indicate that the growth slowdown-mediated increase in the concentration of stable proteins like KinA (Fig 2A) combined with the predicted ultrasensitive dependence of 0A~P on KinA concentration (Fig 2C) is sufficient to explain the observed correlation between growth rate and sporulation decision (Fig 2D). Previous studies have also shown that no signals external to the phosphorelay are essential for KinA to be able to activate sporulation (Eswaramoorthy *et al*, 2011; Devi *et al*, 2015). However, neither of these results is sufficient to exclude the possibility that in addition to these effects, KinA activity is also modulated by an unknown nonessential signal that is correlated with growth rate. To check for this possibility, we need to see whether the amount of KinA required for sporulation would be the same or different at different growth rates. However, in wild-type cells these effects are inseparable because of the one-to-one relationship between KinA level and growth rate under the feedback regulation of phosphorelay (Predich *et al*, 1992; Fujita & Sadaie, 1998).

To circumvent this problem, we can use an engineered strain in which transcription of *kinA* is independent of 0A activity. To this end, we can use a strain in which KinA is externally controlled by

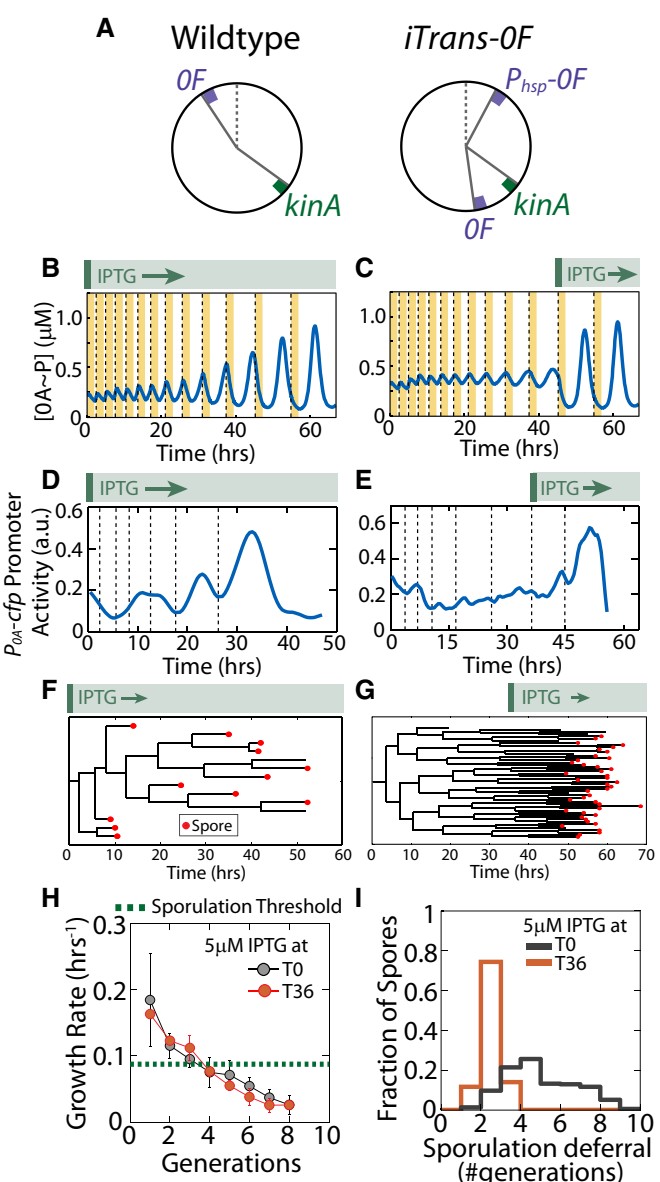

**Figure 4.    Selective exposure to slow growth eliminates sporulation deferral.**

A    Arrangement of *kinA* and *0F* on the chromosome in the wild-type and *iTrans-0F* strains. In the *iTrans-0F* strain, 0A–P pulsing is only observed when the *oriC*-proximal $P_{hsp}$-*0F* is induced with IPTG.

B, C    Timing of induction of the *oriC*-proximal copy of *0F* affects 0A~P pulsing in the *iTrans-0F* strain. Mathematical modeling results predict that if the *oriC*-proximal $P_{hsp}$-*0F* is induced at the start of starvation (B), pulse amplitudes increase gradually over multiple cell cycles similar to WT. If $P_{hsp}$-*0F* is induced later during starvation (C), when growth is slow, 0A~P pulse amplitudes increase immediately without a multi-cell-cycle delay. Yellow bars and dashed lines represent DNA replication periods and cell division, respectively.

D, E    Experimental measurements of $P_{OA}$ activity show that both early and late induction of the *oriC*-proximal $P_{hsp}$-*0F* rescue 0A activity pulsing. These measurements also confirm that early induction leads to a gradual increase in 0A activity pulse amplitudes (D) and late induction leads to an immediate switch from no pulsing to high amplitudes (E).

F, G    Timing of the induction of *0F* induction affects timing of sporulation in the *iTrans-0F* strain. Example lineage trees of the *iTrans-0F* strain show that sporulation timing (spores are marked by red dots) is heterogeneous and unsynchronized when the *oriC*-proximal $P_{hsp}$-*0F* is induced early at T0 (F) and well synchronized when *0F* is induced late at T36 (G).

H    Measurements of growth rate show that slowdown of growth during starvation is noisy but is not affected by whether *0F* is induced early (T0 —gray) or late (T36—orange). Dots and error bars show means and standard deviations of growth rates for each generation, respectively. Note that it takes ~5 generations for the mean growth rate to cross the threshold growth rate below which cells sporulate.

I    Histograms of the number of generations for which cells defer sporulation after *0F* induction in the *iTrans-0F* strain. Gray and orange histograms show sporulation deferral for early (T0) and late (T36) induction, respectively.

Source data are available online for this figure.

inducible promoter. Crucially, simulations of our mathematical model of the phosphorelay show that $0A\sim P$-KinA feedback is not essential for pulsing as long as inducible KinA gene is integrated in the terminus-proximal locus (Fig EV4). Thus, we can use an inducible system to do this test without affecting the pulsatile nature of $0A\sim P$ activation. For each fixed inducer concentration, our model shows that the cellular concentration of KinA increases with decreasing growth rate (similar to the increase in YFP shown in Fig 2A). As a result, for each fixed inducer concentration there will be a KinA threshold and a growth rate threshold for sporulation. Examining how these thresholds depend on the KinA induction level would allow us to test whether KinA activity is also modulated by some unknown starvation signal. Our modeling results predict that if KinA is activated by a growth-rate-correlated signal, different threshold levels of KinA will be required to reach the same

threshold level of $0A\sim P$. Therefore, we expect that KinA threshold would be higher for faster-growing cells (Figs 5A and EV4F) as compared to slower-growing cells. In contrast, if the growth rate only controls KinA concentration, the same threshold level of KinA is expected across all the different growth rates (Figs 5B and EV4G).

Following this rationale, we tested the two hypotheses by using a strain in which a functional KinA-GFP fusion protein is expressed from an IPTG-inducible promoter ($P_{hsp}$). GFP intensity measurements gave us information about single-cell KinA levels, whereas tracking cell length in time-lapse movies allowed us to compute single-cell growth rates. These measurements confirm that KinA levels are indeed sensitive to growth rate and that the resulting growth rate threshold depends on the inducer level (i.e., expression rate from the $P_{hsp}$ promoter; Fig EV5A). Moreover, the growth-dependent

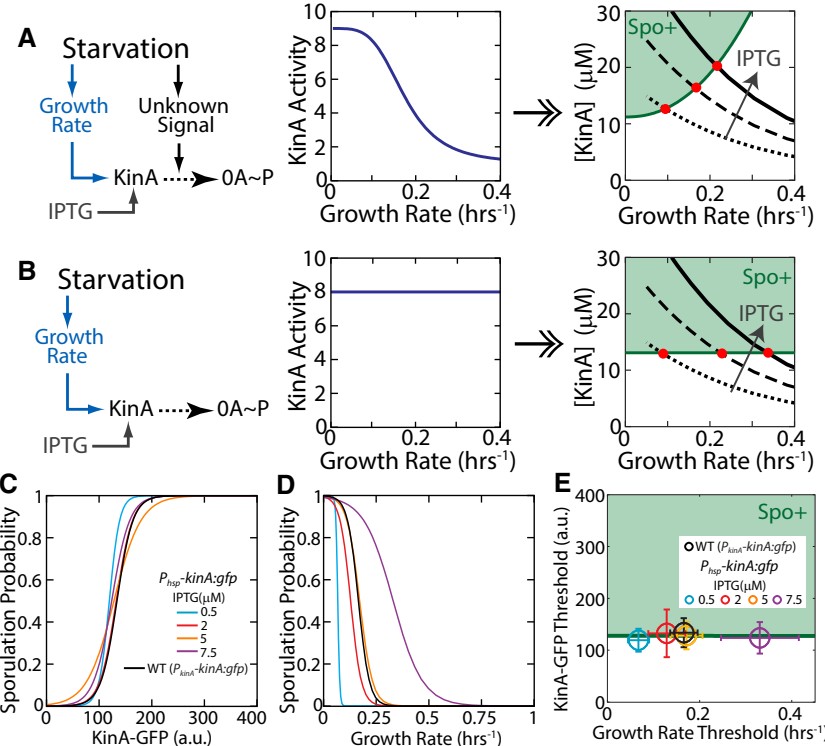

**Figure 5.  Decrease in growth rate affects $0A\sim P$ levels and controls cell fate.**

A    Standard hypothesis for the relationship between starvation, $0A\sim P$, and sporulation cell fate. Starvation affects cell growth and an unknown signal that controls $0A$ activation by KinA. Growth rate and unknown signal level are interrelated since they both depend on starvation. Under this hypothesis, the minimum level of KinA necessary to make cells sporulate is a function of the unknown signal level (middle panel) and by extension the growth rate (right panel). This hypothesis predicts that changes in KinA production rate using an IPTG-inducible system should lead to a change in both the KinA threshold and growth rate threshold for sporulation (red dots).

B    Alternative hypothesis: Signals controlling KinA activity play a negligible role in determining $0A$ activation and sporulation cell fate. Under this hypothesis, the KinA threshold for sporulation is independent of the unknown signal level (middle panel) and by extension the growth rate (right panel). This hypothesis predicts that changes in KinA production rate using an IPTG-inducible system should lead to a change in the growth rate threshold for sporulation but not the KinA threshold (red dots).

C, D  Logistic regression curves indicating the probability of producing a spore as predicted using measurements of KinA-GFP level (C) and growth rate (D).

E    KinA and growth rate thresholds for sporulation in the $P_{hsp}$-KinA and WT strains. KinA and growth rate thresholds were calculated using the results of the logistic regression in (C, D), respectively. Significance of IPTG dependence of KinA and growth thresholds was determined by logistic regression of pooled data ($N_{pooled}$ = 547) and applying a $t$-test to determine whether the regression coefficient for IPTG level is zero (see Materials and Methods for details). Note that the KinA threshold does not depend on IPTG level ($P$-value for IPTG coefficient = 0.70784, $N_{pooled}$ = 547), whereas the growth rate threshold decreases with increasing IPTG ($P$-value for IPTG coefficient = 2.601e-26, $N_{pooled}$ = 547) as predicted by the alternative hypothesis in (B). Error bars show standard errors.

Source data are available online for this figure.

increase in KinA-GFP levels is well matched by model (see equation 1) for protein accumulation.

Next, using the same procedure employed in Fig 2D, we performed logistic regression on the data for the inducible KinA strain to estimate the probability of sporulation as a function of KinA-GFP level or as a function of growth rate. Our results showed an ultrasensitive dependence of the probability of sporulation on both metrics irrespective of the IPTG level used (Figs 5C and D, and EV5B and C). Using these ultrasensitive functions, we identified threshold values for KinA-GFP above which and growth thresholds below which cells sporulate with at least 50% probability (Fig EV5B). Comparing the two types of thresholds for different IPTG concentrations (Fig 5E), we found that the growth rate threshold for sporulation increases as a function of IPTG, whereas the KinA-GFP threshold does not depend on IPTG level. Notably, the severalfold increase in the growth rate threshold as a function of increase in *kinA* expression rate (Fig 5E) indicates that cells are able to sporulate at dramatically different levels of starvation. However, the constant KinA threshold across these different levels of starvation indicates that KinA activity does not change across these conditions.

Thus, our measurements showed that the threshold amount of KinA level necessary to trigger sporulation was not dependent on the level of IPTG. To determine whether the same KinA threshold also controls sporulation in the wild-type strain, we applied the same logistic regression analysis used above to a strain in which a KinA-GFP fusion protein is expressed from its native promoter (Fig EV5). Data for this strain showed an ultrasensitive dependence of probability of sporulation on both KinA-GFP levels and growth rates similar to the inducible strain (Figs 5C and D, and EV5B and C). Further, we found that threshold level of KinA for this wild-type strain is the same as that for the IPTG-inducible strain (Fig 5E).

An alternative explanation for the growth dependence of sporulation is that slowdown of growth leads to accumulation of KinA but does not affect highly unstable negative regulators of KinA like Sda. To examine whether the growth dependence of cell fate results from relative changes in the amounts of KinA and Sda, we constructed an *sda*-deletion strain and tested its 0A activity and growth rate response. As shown in Fig EV6, the sda-deletion strain still shows pulsing and Δsda cells sporulate once cell growth is below a threshold similar to wild-type cells. Comparing the sda-deletion strain to wild type, we found no statistically significant difference in the growth rate threshold required to trigger sporulation. This result demonstrates the limited role of Sda in modulating the growth dependence of sporulation in our conditions.

These results imply that KinA activity is not growth rate dependent in our experimental conditions (Fig 5A) and plays no role in controlling sporulation. Instead, these results show that sporulation is triggered by an increase in KinA concentration due to slowdown in growth rate, but independent of any signals modulating KinA activity (Fig 5B).

## Discussion

Previous studies of *B. subtilis* sporulation have primarily focused on identifying specific environmental and metabolic signal molecules that regulate the phosphorelay response. Here, taking a different approach, we have focused on the role of cell growth in determining

the phosphorelay response and found that it actually plays a major part in controlling the response to starvation and the sporulation decision.

Our results show that the amplitude of 0A~P pulses increases as cell growth rate decreases. This inverse dependence can be mechanistically understood using two observations about the phosphorelay network controlling 0A~P: (i) The concentration of the cytosolic kinase KinA is the rate-limiting factor that determines 0A~P pulse amplitude (Eswaramoorthy *et al*, 2010a; Levine *et al*, 2012; Narula *et al*, 2012) and (ii) the concentration of KinA, a stable protein, increases as growth slows down. Here, by using a mathematical model to study the combined effect of these two, we were able to demonstrate how slowdown of growth leads to accumulation of KinA which in turn increases 0A~P amplitudes and consequently sporulation. Crucially, our results show that growth slowdown is unaffected in the absence of pulsing which suggests that growth rate does not depend on 0A~P levels. Moreover, since a 0A~P threshold is known to control sporulation cell fate (Fujita *et al*, 2005; Narula *et al*, 2012), we found that there is a corresponding threshold growth rate below which cells sporulate. This growth threshold depends on the KinA production rate and can be increased severalfold with an increase in *kinA* expression rate. Nevertheless, the same level of KinA triggers sporulation regardless of conditions. Taken together, these results suggest that not only does cell growth play a major role in determining 0A activity and cell fate, but it might be the primary signal by which cells gauge their starvation level.

This role of growth in determining cell fate offers fresh insight into the variability of sporulation timing during starvation. Several recent reports have shown that *B. subtilis* cells can defer sporulation for multiple generations during starvation and that this deferral period is highly heterogeneous (Chastanet *et al*, 2010; de Jong *et al*, 2010; Levine *et al*, 2012). In agreement with these reports, our results show that sister cells frequently adopt different cell fates during starvation (Fig 3A). It has been suggested that this heterogeneity is the result of the stochasticity of gene expression. However, our results show that while gene expression stochasticity may play a role, the heterogeneity of sporulation timing can be largely explained by the variability of cell growth rates during starvation. We have shown that the gradual but noisy decrease in growth rate during starvation is the reason sporulation is delayed. Perturbation of the dynamics of growth rate slowdown can increase or decrease the deferral of sporulation. Moreover, the selective exposure of the phosphorelay to slow growth eliminates the deferral entirely.

The reliability of growth rate as a predictor of cell fate also calls into question previous suggestions that the phosphorelay is a "noise generator" that creates heterogeneity in 0A activity levels and sporulation timing as part of a bet-hedging strategy (Chastanet *et al*, 2010). Instead, our results suggest that at least in our conditions, heterogeneity in 0A activity, cell fate, and sporulation timing is the result of growth rate variability, that is, the noise is generated upstream of the phosphorelay. Why growth rates are so variable in starvation conditions remains an open question, but the fact that growth rates of sister cells are only weakly correlated (Fig EV3B) suggests that some sort of intrinsic noise is the driver of this heterogeneity. Notably, it has been recently shown that stochastic fluctuations in concentrations of rate-limiting metabolic enzymes can result in growth rate heterogeneity (Kiviet *et al*, 2014). A similar mechanism may be responsible for the growth

heterogeneity in our experiments; however, this is a subject for future studies.

A key open question in the field of *Bacillus subtilis* sporulation that we address here is: How do cells sense and integrate the environmental cues that trigger spore formation? Previous studies have proposed that starvation signals control sporulation by acting on three PAS repeats in the KinA N-terminal sensor domain to control KinA autophosphorylation (Stephenson & Hoch, 2001; Wang *et al*, 2001; Kolodkin-Gal *et al*, 2013). However, more recent studies suggest that the major role of the N-terminal PAS domain is to form a stable tetramer as an active form and the kinase is constitutively active regardless of culture conditions (Eswaramoorthy *et al*, 2009, 2011; Eswaramoorthy & Fujita, 2010). Moreover, as shown previously (Eswaramoorthy *et al*, 2010a; Narula *et al*, 2012) and recapitulated by our results here, KinA induction can override starvation requirements and force even cells that are not starving to sporulate. These results together with our observation that the KinA concentration necessary to trigger sporulation does not depend on growth rate (Fig 5E), indicate that at least in our experimental conditions KinA activity is mainly unmodulated and that growth-dependent increase in KinA is the primary signal for sporulation.

These results clearly establish the key role played by cell growth rate in determining the starvation response. However, we can not rule out that, under different conditions, other signals may also affect the phosphorelay. Indeed, several proteins like KipI, Sda, Rap, BmrD, and Obg have been shown to respond to metabolic, environmental, cell density, and cell-cycle-related signals and modulate the phosphorelay response (Vidwans *et al*, 1995; Perego & Hoch, 1996; Wang *et al*, 1997; Perego, 1998; Burkholder *et al*, 2001; Fukushima *et al*, 2006). It will be interesting to investigate how these regulators impact the growth dependence of sporulation cell fate, that is, do they act as modulators that control the growth threshold or do they act as all-or-none checkpoint mechanisms that prevent any growth-dependent increase in 0A activity and sporulation.

Notably, several of the above-mentioned proteins are negative regulators of sporulation that act on either KinA or Spo0F and suppress the phosphate flux in the phosphorelay. This suggests that they may act as essential checkpoints that prevent sporulation under conditions where growth rate is low despite the presence of nutrients, for example, at low temperatures or in the presence of stress agents like antibiotics and ethanol. Such checkpoint regulators may also play an important role in preventing slow-growing cells from sporulating in conditions like MSgg media where *Bacillus subtilis* needs to follow alternate differentiation programs like biofilm formation (Chai *et al*, 2008).

In contrast to the negative regulators discussed above, a recent report has postulated the existence of a positive regulator of sporulation called extracellular factor FacX (Ababneh *et al*, 2015). Its existence was demonstrated by triggering sporulation via IPTG-controlled KinA induction in exponentially growing culture. Notably, Ababneh *et al* showed that induction of KinA leads to sporulation in exponentially growing cells only if the medium from stationary phase cultures is added. This suggests that the extracellular factor FacX, which is produced during post-exponential growth, can create a permissive environment for sporulation during exponential growth. Crucially, addition of FacX to exponentially growing cultures does not trigger

sporulation without KinA induction. This suggests that the growth rate requirement and other regulatory mechanisms remain active in the presence of FacX and prevent sporulation. As the molecular identity of FacX remains unknown, it is unclear how it triggers sporulation or what controls FacX production. However, since its accumulation is only significant in the post-exponential phase, it will be interesting to determine whether its production is growth rate dependent.

In summary, our results reveal a novel decision strategy based on the pulsatile 0A~P response to starvation: The phosphorelay only responds with a pulse upon completion of DNA replication, and the amplitude of each pulse encodes the cell's growth rate which is an indicator of the extent of starvation. This simple strategy allows cells to defer commitment to sporulation for as long as environmental conditions remain conducive to growth while bypassing the need for specific metabolite sensing. Notably, the sensitivity of stable protein levels to cell growth that enables this decision strategy is a universal feature of bacterial physiology and not unique to the sporulation network. As a result, similar growth rate-dependent strategies for controlling the starvation response could very well be employed in a wide range of other systems.

# Materials and Methods

### Strain construction

Appendix Table A1 lists *Bacillus subtilis* strains used in this study. All strains are isogenic to *B. subtilis* PY79.

#### iTrans-0F strain construction

For the inducible *0F* cassette, the *spo0F* gene was PCR-amplified from *B. subtilis* PY79 with the addition of optimal RBS and linker (AAGGAGGAAAGTCACATT) and including 43-bp fragment downstream. It was ligated to a derivative of PLD30, JDE131 plasmid (Sp$^R$) next to the IPTG-inducible $P_{hyperspank}$ promoter (between HindIII and NheI restriction sites. This cassette was transformed into AmyE locus of a reporter *B. subtilis* strain constructed previously (Kuchina *et al*, 2011b) harboring PspoIIR-YFP in *SacA* locus and Pspo0A-CFP on PHP13 low-copy plasmid. Then, we knocked out native *spo0F* in this strain. For the 0F deletion construct, the 5′ and 3′ fragments of *0F*-adjacent genomic sequence were PCR-amplified using the following primers: *GAGGCGCC*CCTGTCGCTTTCTGTCACTTCCTCAG and *TCGAATTC*GCAAAATACGAATGCCGTATTGATCATCAACGA for 5′ arm, and *GATCTAGA*GACATCGACGAAATCAGAGACGCCGTCAAAAAATATCTGCCCCTGAAGTCTAAC and *TCGTCGAC*CCTTCGGAAACACCAAGGATCACTGGAG for 3′ arm, and introduced into per449 (Kan$^R$) vector between XbaI and SalI restriction sites. The specific native genomic Spo0F knockdown in the resulting *i0F$^{amyE}$* strain was confirmed by PCR and sequencing. Finally, the native promoter *spo0F*-rescue cassette was constructed and introduced as follows. Gene *spo0F* with its native $P_{spo0F}$ promoter (165 bp upstream) and 43-bp downstream sequence were PCR-amplified from *B. subtilis* PY79 with the addition of a terminator upstream (CCAGAAAGTCAAAAGCCTCCGACCG) and ligated between BHI and XbaI restriction sites to ECE173 (Pm$^R$) integrating into GltA locus. This construct was transformed into the *i0F$^{amyE}$* strain resulting in *iTrans-0F* strain.

*Other reporter strains, cloning vectors, and transformation method*
The reporter strains for *spo0A* and *spo0F* promoters and the $P_{kinA}$-KinA-GFP and $P_{hsp}$-KinA-GFP strains were described previously (Fujita & Losick, 2005; Kuchina *et al*, 2011b). The strain expressing YFP from the IPTG-inducible $P_{hsp}$ promoter was constructed and characterized for dose response earlier (Süel *et al*, 2007). To identify DNA replication periods in time-lapse experiments (Fig EV1I), we used a strain described previously that expresses a fluorescent DnaN-YFP fusion protein from the IPTG-inducible $P_{hsp}$ promoter (Veening *et al*, 2009; Su'etsugu & Errington, 2011; Narula *et al*, 2015).

To facilitate segmentation in the glucose addition experiments (Fig 3), we transformed a plasmid bearing a constitutively expressed promoter $P_{rpsD}$ driving mCherry expression (constructed as described in Zhang *et al*, 2015) into the dual sporulation reporter strain harboring $P_{spoIIR}$-YFP at SacA genomic locus and $P_{spo0A}$-CFP expressed from the low-copy PHP13 plasmid, also described earlier [(Kuchina *et al*, 2011b), "0A (5x)-IIR"]. Promoter sequences were defined as follows: PrpsD—chromosomal sequence 2853257 to 2853567; Pspo0A—chromosomal sequence 2518060 to 2518350; and PspoIIR—chromosomal sequence 3794404 to 3794543 (Kuchina *et al*, 2011a,b).

The vectors used in this study are as follows: ECE174, integrating into the *sacA* locus (constructed by R. Middleton and obtained from the Bacillus Genetic Stock Center); pLD30, designed to integrate into the *amyE* locus (kind gift from Jonathan Dworkin, Columbia University); ECE173, designed to integrate into the *gltA* locus (Middleton & Hofmeister, 2004) (constructed by R. Middleton and obtained from the Bacillus Genetic Stock Center); per449, a generic integration vector constructed for integration into the gene of interest (kind gift from Wade Winkler, UT Southwestern); and the bifunctional cloning plasmid pHP13 carrying the replication origin of the cryptic *B. subtilis* plasmid pTA1060 (Haima *et al*, 1987). One-step *B. subtilis* transformation protocol was followed.

**Culture preparation and microscopy**

*Culture preparation*
For imaging, *B. subtilis* culture was started from an overnight LB agar plate containing appropriate antibiotics (final concentrations: 5 μg/ml chloramphenicol, 5 μg/ml neomycin, 5 μg/ml erythromycin, 5 μg/ml phleomycin, and 100 μg/ml spectinomycin). Strains containing multiple resistance genes were grown on a combination of no more than three antibiotics at a time. Cells were resuspended in casein hydrolysate (CH) medium (Sterlini & Mandelstam, 1969) and grown at 37°C with shaking. After reaching OD 1.8–2.0, cells were washed once and resuspended in 0.5 volume of resuspension medium (RM) (Sterlini & Mandelstam, 1969). The resuspended cells were grown at 37°C for 1 h, then diluted 15-fold, and applied onto a 1.5% low-melting agarose pad made with RM-MOPS medium with desired IPTG or glucose concentration, if necessary. The pads were covered, left to air-dry for 1 h at 37°C, and placed into a coverslip-bottom Willco dish for imaging. For the late IPTG addition experiment, a small drop (5 μl) of IPTG dissolved in RM was applied onto the RM-MOPS pad between image acquisitions through an opening in the dish lid.

*Time-lapse microscopy*
Differentiation of *B. subtilis* microcolonies was monitored with fluorescence time-lapse microscopy at 37°C with an Olympus IX-81 inverted microscope with a motorized stage (ASI) and an incubation chamber. Image acquisition was set to every 20 min with a Hamamatsu ORCA-ER camera. Custom Visual Basic software in combination with the Image Pro Plus (Media Cybernetics) was used to automate image acquisition and microscope control.

*Image analysis*
A combination of custom written MATLAB programs, Microbe-Tracker (Sliusarenko *et al*, 2011), and freely available ImageJ plug-ins (Rasband, W.S., ImageJ, U. S. National Institutes of Health, Bethesda, Maryland, USA, http://imagej.nih.gov/ij/, 1997–2014) was used to analyze microscopy data as described below.

**Data analysis**

*Quantification of cell growth rates*
The mean cell growth rate for individual cell cycles (Figs 1C and F, 2A and D, 3C and F, 4F, EV1A, D, G and H, EV3, EV5A, EV6A and D, and Appendix Fig S1) was quantified using the measurements of cell length in the time-lapse data. We first calculated the instantaneous cell growth rate at every frame as:

$$\mu(t) = \frac{1}{L(t)}\frac{dL(t)}{dt} = \frac{d\log L(t)}{dt} = \frac{\log(L(t + \Delta t)) - \log(L(t))}{\Delta t}$$

where $L(t)$ is the cell length at time $t$ and $\Delta t$ is the time difference between successive frames (20 min). For cell cycles that result in vegetative division, mean growth rate during a cell cycle was defined as the average of $\mu(t)$ over the cell-cycle duration. For cell cycles that end in sporulation, the mean growth rate was the average of $\mu(t)$ over the cell-cycle duration until the asymmetric division. Depending on the strain, the asymmetric division was defined either as $P_{spoIIR}$ activation (for the strains in which a $P_{spoIIR}$ reporter was present) or as the time frame 2 h before the appearance of the phase-bright forespore.

*Calculation of promoter activity*
The measurements of promoter activities for $P_{0A}$-*cfp/yfp* promoter (Figs 1E and F, 2C and D, and EV6C and D) $P_{0F}$-*yfp* promoter (Fig EV1C and G) and $P_{hsp}$-*yfp* promoter (Fig EV1F) refer to rate of protein production calculated from fluorescence time-lapse data using the same procedure as previously reported (Narula *et al*, 2015).

*Quantification and characterization of promoter activity pulses*
Promoter activity time series determined from fluorescence microscopy were smoothed using the MATLAB *smooth* function by employing a Savitzky–Golay filter with a third-order polynomial over a sliding window of five frames. After smoothing, the maximum promoter activity during the cell cycle was used as the peak promoter activity (Figs 2C and D, EV1G, and EV6D).

*Growth rate dependence of induced gene expression from $P_{hyperspank}$ promoter*
The growth dependence of fluorescent protein levels in $P_{hsp}$-*yfp* and $P_{hsp}$-*kinA-gfp* was modeled with equation 1 (Figs 2A and EV5A). In

this equation, $V(\mu) = AL(\mu)$ where A and $L(\mu)$ are the cross-sectional cell area and cell length, respectively. The cross-sectional area was assumed to be fixed and growth independent. The dependence of $L(\mu)$ on growth was determined by fitting the data for change in cell length at division as a function of growth rate (Fig EV1H). We used the following phenomenological expression for $L(\mu)$: $L(\mu) = 3.466*\exp(-0.689/\mu) + 3.743$ μm.

To isolate the growth dependence of gene expression in the $P_{hsp}$-*yfp* expressing strain, the fluorescence measurements were normalized by the concentration predicted by the fitted equation at $\mu = \log(2)/h$.

*Estimation of the growth rate, OA activity, and KinA thresholds for sporulation*

The thresholds for predicting sporulation cell fate (Figs 2D and 5C, D and E) were determined independently for each strain (and in the inducible KinA strain for each IPTG level). In every case, observations for individual cell cycles included the independent variables: (i) the mean cell-cycle growth rate and (ii) either the peak $P_{0A}$ promoter activity (Fig 2D) or the peak KinA-GFP level during the cell cycle (Fig 5C and D); and the dependent variable: cell fate (i.e. vegetative division = 0 or sporulation = 1). These observations were fit with a logistic function using the MATLAB *glmfit* function to predict the probability of sporulation as a function of each of the individual independent variables (Figs 5C and D, EV2 and EV5B and C). Each outcome of the dependent variable, cell fate, was assumed to be generated from a binomial distribution for the logistic regression. 95% confidence intervals of the logistic regression curves (Figs EV2 and EV5B and C) were also calculated using the *glmfit* function.

Threshold values of the independent variables were defined as the value at which the fitted logistic regression predicts probability of sporulation to be equal to 0.5. The *glmfit* function was also used to calculate the standard error of the thresholds. For the growth rate threshold, cell cycles with growth rate greater than the threshold were classified as vegetative and those with growth rate lower than or equal to the threshold were classified as sporulating. For the peak $P_{0A}$ promoter activity or the KinA-GFP thresholds, cell cycles with values greater than the threshold were classified as sporulating. These threshold values were then used to predict cell fates and the calculate three performance measures: (i) false-positive rate (FPR, fraction of vegetative cells that were incorrectly predicted to sporulate), (ii) true-positive rate (TPR, fraction of sporulating cells that were correctly predicted to sporulate), and (iii) total error rate (fraction of total cells whose cell fate was incorrectly predicted).

To determine the effectiveness of each threshold as a cell-fate prediction method, we computed the receiver operating characteristic (ROC) curve for each case (Figs 2E and EV5D). This ROC curve is a commonly used way to characterize the performance of binary classifier in signal detection theory (Provost & Fawcett, 2001). We compute it by varying the threshold and calculating the false-positive rate and true-positive rate as functions of threshold value (Provost & Fawcett, 2001). Next, we computed the AUC (area under the ROC curve) to estimate the performance of growth rate, peak $P_{0A}$ promoter activity, and KinA-GFP level threshold based on cell-fate predictions for the various strains. The computed AUCs and optimal thresholds for each case are summarized in Fig EV5D. The

results indicate that all three variables can be used to robustly predict sporulation cell fate for the WT and inducible KinA strains.

*Quantification of growth rate and sporulation fraction dynamics*

To calculate the growth and sporulation fraction dynamics based on generations (Fig 3), first we binned the observations for individual cell cycles of WT cells by generation number (the cell cycle at the start of the time-lapse experiment was labeled generation 0). The fraction of cell cycles in each generation bin that ended in sporulation were used to calculate the experimental sporulation fraction dynamics (red dots in Fig 3B and E). Next, the WT growth rate threshold calculated from the data in Fig 2D was used to classify the cell cycles as vegetative (growth rate > threshold) or sporulating (growth rate ≤ threshold). The ratio of the number of cell cycles classified as sporulating to the total number of cell cycles in each generation bin was used as the predicted sporulating fraction (black curves in Fig 3B and E).

To calculate the growth dynamics based on time (Appendix Fig S1), first we divided the whole time span of the time-lapse experiment into 2-h time bins. Next, for the each cell cycle in the experiment, its mean cell-cycle growth rate was calculated in all 2-h time bins that were spanned by that cell cycle. The mean and standard deviations of cell-cycle growth rates in each time bin were calculated to determine the colony growth rate dynamics. These results were then used with a population dynamics model to explain how nutrient addition affects the growth slowdown dynamics (Appendix Fig S1).

**Statistical analysis**

Significance of correlation between "peak promoter activity" and "growth rate" variables (Figs 1F and EV1G) was tested using the standard two-tailed *Z*-test at the 0.05 confidence level. Sample sizes under the null hypothesis of no correlation were chosen to ensure that a correlation coefficient of 0.2 could be detected with a power of 90%.

To determine whether growth rate or KinA-GFP thresholds for sporulation depend on IPTG in the inducible KinA strain (Figs 5E and EV5E and F), we extended our logistic regression approach and included IPTG as a predictor for cell fate. Specifically, data for growth rates, KinA-GFP level, and cell fate for IPTG induction levels and WT cells was pooled ($N_{pooled} = 547$) and fit with a logistic regression function of the form: $\text{logit}(y) \sim \beta_0 + \beta_1*x_1 + \beta_2*x_2$, where $y$ = cell fate, $x_1$ = either growth rate or KinA-GFP level, $x_2$ = IPTG indicator variable (= IPTG concentration for inducible KinA and 0 for WT), and $\beta_{0-2}$ = regression coefficients. The *fitglm* function in MATLAB was used to fit the pooled data with this model. The same function was used for each regression coefficient to also calculate the *t*-statistic for a test that the coefficient is zero and corresponding *P*-value. A $\beta_2$ regression coefficient that is significantly different from zero was taken to indicate dependence of the sporulation threshold on IPTG. Using this procedure, we found the following *P*-values for growth rate dependence: 0.030967 ($\beta_0$), 4.8959e-23 ($\beta_1$), 2.601e-26($\beta_2$), thus indicating that growth threshold depends on IPTG. In contrast, we found the following *P*-values for KinA-GFP level dependence: 8.1272e-30 ($\beta_0$), 1.2117e-24 ($\beta_1$), 0.70784 ($\beta_2$), thus indicating that KinA threshold does not depend on IPTG.

                                                        

The same procedure was used to determine whether the growth rate or 0A activity thresholds varied between WT and Δsda cells (Fig EV6E and F). Data for growth rates, peak $P_{0A}$ promoter activity, and cell fate for WT and Δsda cells was pooled ($N_{pooled} = 449$) and fit with a logistic regression function of the form: logit($y$) ~ $\beta_0 + \beta_1 * x_1 + \beta_2 * x_2$, where $y$ = cell fate, $x_1$ = either growth rate or peak $P_{0A}$ promoter activity, $x_2$ = strain indicator variable (= 0 for WT and 1 for Δsda), and $\beta_{0-2}$ = regression coefficients. Using this model and the *fitglm* function, we found the following *P*-values for growth rate dependence: 3.2012e-20 ($\beta_0$), 8.4018e-23 ($\beta_1$), 0.17138 ($\beta_2$), thus indicating that growth threshold does not depend on whether the strain is WT or Δsda. We also found the following *P*-values for peak $P_{0A}$ promoter activity dependence: 2.8499e-23 ($\beta_0$), 2.131e-29 ($\beta_1$), 0.41013 ($\beta_2$), thus indicating that peak $P_{0A}$ promoter activity threshold also does not depend on whether the strain is WT or Δsda.

**Expanded View** for this article is available online.

## Acknowledgements

This work is supported by National Science Foundation grants MCB-1244135 (to OAI), EAGER-1450867 (to GS), MCB-1244423 (to MF), National Institute of Health NIGMS grant R01 GM088428 and the San Diego Center for Systems Biology (NIH Grant P50 GM085764) (to GS), and HHMI International Student Fellowship to JN.

## Author contributions

JN, AK, GMS, and OAI designed the research. JN developed the mathematical model. AK, MF, and FZ constructed the strains and performed the experiments. JN and AK analyzed the data. All authors wrote the paper.

## Conflict of interest

The authors declare that they have no conflict of interest.

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
