## [Review Process File · Molecular Systems Biology]

Slowdown of growth controls cellular differentiation

Jatin Narula, Anna Kuchina, Fang Zhang, Masaya Fujita, Gurol M. Suel and Oleg A. Igoshin

Corresponding author: Oleg A. Igoshin, Rice University

Review timeline:

Submission date:	12 November 2015
Editorial Decision:	10 January 2016
Revision received:	20 March 2016
Accepted:	07 April 2016

Editor: Thomas Lemberger

Transaction Report:

1st Editorial Decision

10 January 2016

We have now heard back from the two referees who agreed to evaluate your manuscript. As you will see from the reports below, the referees find the topic of your study of potential interest. They raise, however, several points that should be convincingly addressed in a revision of this work.

The reviewers point to the need to avoid overclaiming the conclusions of the work and to perform key experiments in *sda* mutants to evaluate the contribution of this mechanism as compared to slowdown.

Reviewer #1:

Referee report on 'Slowdown of growth controls cellular differentiation'.

Narula and co-workers use a combination of single cell analysis of wild type and mutants and mathematical modeling to pinpoint the cues for initiation of sporulation in *Bacillus subtilis*. It is shown that sporulation is correlated with slow growth. This correlation can surprisingly explain many features of the sporulation dynamics and suggests that complicated sensing mechanisms are not really essential for effective sporulation. Simply growing slow will increase the concentration of the primary sporulation kinase KinA which then switches on the unidirectional cascade leading to endospore formation. This work recapitulates a couple of previously made observations and provides a unifying model for sporulation initiation. Specifically, this work shows that KinA does not require a specific (biochemical) signal to be active. This was known but now explains that increasing KinA concentration will lead to a higher probability of entering sporulation. Since KinA (or any stable protein) will be increased in its concentration simply by reduced growth (and dilution) this explains the correlation between slow growth and Spo0A activity.

In general, this is an interesting study making good use of single cell analysis and mathematical modeling. The growth-rate correlation with sporulation is novel and important although it is not yet clear why some cells grow slower than others (and thus sporulate or not).

See below my comments and questions on this work:

1. L13, how is Spo0A-activity correlated with growth rate? Slower growthhigher Spo0A activity. This should be made clear here.
2. L67, I believe it was Veening et al. 2009 G&D who first observed pulsatile behavior in Spo0A activity.
3. Figure 1, From the legend and text it is not clear if we are looking at the trace of a single cell, or at an average trace of many cells. Statistics on the number of analysed cells is missing.
4. Is the growth rate also inherited to the next generation? In other words, if a cell grows slow but still divides, will both daughter cells grow slow (and thus both have a higher probability to enter spore formation)? This should be analysed.
5. L93, increased amplitude that is coordinated with a decrease..
6. P7L148, L383, Fig. 4. A known negative regulator of KinA activity is Sda. Sda was shown to be highly unstable (half-life of about 5 min) (Burkholder et al. 2001 Cell). Since a decrease in growth rate leads to accumulation of stable proteins such as KinA, reduced growth will also lead to increased activity of KinA because Sda is unstable. However, the authors conclude that additional regulators such as Sda only play a minor role (in the Discussion). The authors should perform their experiment also in an sda mutant. On basis of the literature and the here observed growth rate dependent KinA enrichment phenomenon, one would predict that the pulse amplitude would be greater and the probability to sporulate would increase. On the basis of the current text one would not expect any effect of the sda mutant. It will be very interesting to see the result.
7. L189, increased 0A-P amplitudes
8. L221, this was already shown in Chung et al. 1994 JBact.
9. Figure 2. The authors need to provide some single cell traces of the Phsp-YFP strain. Does this also oscillate under the here-used experimental conditions?
10. L280, bacteria do not choose, they can't think.
11. Can the authors speculate why there is so much heterogeneity in growth rates in these otherwise isogenic cells grown under more or less identical conditions?
12. Is growth and sporulation heterogeneity observed under more uniform conditions such as within microfluidic devices that can keep the nutrient, oxygen and waste flow uniform?
13. L454, This was a very complicated way to show that KinA does not require an external signal to activate sporulation. Here it should be noted that similar conclusions have been made previously by the authors (e.g. Devi et al, 2015 JBact, Eswaramoorthy Jbact).
14. L509, this statement is incorrect. It has not been shown that bet-hedging or memory controls sporulation. Rather, it has been suggested that heterogeneity in spore formation might provide a bet-hedging strategy (see e.g. de Jong et al. 2011 Bioessays). The here-discovered growth-rate heterogeneity driving the sporulation decision might still be part of an evolutionary bet-hedging strategy. Also, the authors have not tested whether the slow growth phenotype can be inherited (see my point 4).

Reviewer #2:

This report by Naruta et al is the latest in a fascinating series of studies from the same group that have shed much light on the complex regulation of sporulation in *B. subtilis*. The authors have used a combination of modeling and wet experiments to support an enticingly simple model for the crucial signal for the onset of spore formation. It is suggested that starvation does not result in a chemical signal that activates the phosphorelay, but that the slowdown in cell growth that results from starvation automatically boosts the concentration of key regulatory proteins, causing the accumulation of Spo0A~P, the master activator of spore transcription. The authors propose that this is the "primary signal for sporulation" and that all else is just a matter of "checkpoints and fine tuning".

The data in this report are ingenious and convincing and this paper should have a major impact. But there are a few points that are either confusing to me or perhaps merit discussion. I would argue that although the authors have made strong case, it has been unnecessarily overstated.

1. The Herman lab has recently published a paper in PLOS One that uses the artificial induction of KinA to show that the initiation of spore formation cannot proceed when the culture is growing exponentially, but does take place when growth slows. This is completely consistent with the model advanced from the present study. However, when conditioned medium is added, containing a so-called factor X, cells in an exponentially growing culture can proceed at least through engulfment. This would seem to be contrary to the expectations from the present study. I realize that the Herman paper does not examine the rate of increase of cell volume as sporulation initiates, but rather the ensemble rate of change of OD. But I think this apparent contradiction deserves comment.

2. It is well known that certain media favor sporulation and others do not. For example, the addition of glucose sharply reduces sporulation. And other media lacking glucose apparently do not allow much sporulation immediately after cells enter stationary phase. One such example is MSgg, commonly used to study biofilm formation. Doesn't this suggest that there are important nutritional signals that are independent of growth rate? Again, I think comment is needed.

3. I find the following to be confusing. On lines 324 and following, the authors describe experiments with their iTrans-0F strain. It is stated on line 330-333 that when IPTG is omitted the gene dosage ratio of kinA/0F is "1" and that as a result, pulses of 0A~P do not occur as a result of DNA replication. This is understandable. But the reason for the pulses is that when the ratio is 0.5, as in the wild-type strain before the completion of replication, the excess of 0F inhibits the phosphorylation of KinA. So, in the iTrans-0F strain growing without IPTG, KinA should not be inhibited. Consequently, KinA~P should be formed continuously at a high rate and sporulation should take place as growth slows. Why is it stated that the strain will not sporulate without IPTG?

4. On lines 525 and following it is stated that knocking out the genes listed on line 518 does not significantly affect sporulation. This is to counter the potential argument that the growth rate model is an over-simplification. However, although eliminating KipA does not inhibit sporulation, the elimination of KipA, a KipI antagonist, does reduce spore formation. This pathway is thought to respond to the availability of nitrogen. Sda inhibits sporulation when replication initiation is inhibited; there is a DNA replication checkpoint. PhrA is but one anti-Rap peptide. When the permease for these peptides is eliminated, sporulation is impaired. So quorum sensing, aside from Herman's FacX, is an important input. True, a yheH knockout is not reduced in sporulation, but yheI/yheH overexpression does lower spore frequency. Could this hint at a role for this ABC transporter in sensing an unknown environmental signal? I think that the growth rate model is quite pretty but we must be careful not to be seduced into embracing an oversimplification. A decision as drastic for the cell as embarking on spore formation is likely to be hemmed in by all sorts of positive and negative signals in addition to decreasing growth rate.

1st Revision - authors' response

20 March 2016

Point-by-point response

Reviewer #1: Narula and co-workers use a combination of single cell analysis of wild type and mutants and mathematical modeling to pinpoint the cues for initiation of sporulation in Bacillus subtilis. It is shown that sporulation is correlated with slow growth. This correlation can surprisingly explain many features of the sporulation dynamics and suggests that complicated sensing mechanisms are not really essential for effective sporulation. Simply growing slow will increase the concentration of the primary sporulation kinase KinA which then switches on the unidirectional cascade leading to endospore formation. This work recapitulates a couple of previously made observations and provides a unifying model for sporulation initiation. Specifically, this work shows that KinA does not require a specific (biochemical) signal to be active. This was known but now explains that increasing KinA concentration will lead to a higher probability of

entering sporulation. Since *KinA* (or any stable protein) will be increased in its concentration simply by reduced growth (and dilution) this explains the correlation between slow growth and *Spo0A* activity.

In general, this is an interesting study making good use of single cell analysis and mathematical modeling. The growth-rate correlation with sporulation is novel and important although it is not yet clear why some cells grow slower than others (and thus sporulate or not).

-We agree with the overall assessment but would like to point out that as indicated below our claim is stronger than “KinA does not require a specific (biochemical) signal”. We show that under our conditions, KinA activity is not modulated as cells starve for longer and slow-down their growth (Please also see response to question 13).

See below my comments and questions on this work:

1. L13, how is *Spo0A*-activity correlated with growth rate? Slower growthhigher *Spo0A* activity. This should be made clear here.

-Rephrased (see line 15).

2. L67, I believe it was Veening et al. 2009 G&D who first observed pulsatile behavior in *Spo0A* activity.

-We've included the reference to Veening et al (line 68). However, it is worth pointing out that their proposed mechanism of pulsing contradicts (1) the evidence in Narula et al., Cell 2015 (translocation of *spo0F* abolished pulsing), (2) results in Fig. S7 that *sda* deletion does not abolish pulsing and (3) the results of Levine et al., PLoS Biology, 2012 that show that *sda* deletion does not abolish pulsing.

3. Figure 1, From the legend and text it is not clear if we are looking at the trace of a single cell, or at an average trace of many cells. Statistics on the number of analysed cells is missing.

-We've made it clear that traces in Fig. 1C-E correspond to a single cell traced for multiple generations (see line 94) whereas the aggregate data in Fig. 1F corresponds to 307 total cell cycles (see line 103).

4. Is the growth rate also inherited to the next generation? In other words, if a cell grows slow but still divides, will both daughter cells grow slow (and thus both have a higher probability to enter spore formation)? This should be analysed.

-Following the reviewer's suggestion, we have analyzed the relationship between growth rates of mother-daughter pairs. As shown in Fig. EV3A, mother and daughter growth rates are weakly correlated ($\rho=0.41$, $p\text{-val}=4.5e-14$, $N_{\text{pair}}=312$) suggesting that slow growing mothers do produce slow growing daughters that are likely to sporulate. However given that the average growth rate decreases every generation, the effect of growth rate heritability on probability of sporulation is difficult to separate from the effects of growth slow-down due to nutrient depletion. We've decided that the best way to address this concern is to compute the correlation between sister cell growth rates. Fig. EV3B shows that this correlation is also quite weak ($\rho=0.36$, $p\text{-val}=4e-06$, $N_{\text{pair}}=156$) indicating that there is a large chance that only one of two daughter cells will sporulate. In fact, many instances of such asymmetry can be seen on Fig. 3A and Fig. 4FG. Thus the heterogeneity of growth dominates the effect of heritability in determining sporulation cell-fate. We have described and discussed these results in the text (see lines 243-254, lines 557-564).

5. L93, increased amplitude that is coordinated with a decrease..

-Corrected (see line 97).

6. P7L148, L383, Fig. 4. A known negative regulator of *KinA* activity is *Sda*. *Sda* was shown to be highly unstable (half-live of about 5 min) (Burkholder et al. 2001 Cell). Since a decrease in growth rate leads to accumulation of stable proteins such as *KinA*, reduced growth will also lead to

increased activity of KinA because Sda is unstable. However, the authors conclude that additional regulators such as Sda only play a minor role (in the Discussion). The authors should perform their experiment also in an sda mutant. On basis of the literature and the here observed growth rate dependent KinA enrichment phenomenon, one would predict that the pulse amplitude would be greater and the probability to sporulate would increase. On the basis of the current text one would not expect any effect of the sda mutant. It will be very interesting to see the result.

-The suggested experiments were performed and now included in the Fig. EV6. Notably, sda-deletion strain still shows pulsing and *Asda* cells sporulate once cell growth is below a threshold. Comparing the sda-deletion strain to wt, we found no statistically significant difference in 0A activity or growth rate thresholds required to trigger sporulation. If anything, the growth threshold is slightly lower in the sda-mutant, however, since this decrease is not statistically significant we've decided not to pursue this point further. (See lines 475-484)

7. L189, increased 0A-P amplitudes

-Corrected (see line 196).

8. L221, this was already shown in Chung et al. 1994 JBact.

-We thank the reviewer for bringing this to our attention. We've included the citation to this reference (see line 228).

9. Figure 2. The authors need to provide some single cell traces of the Phsp-YFP strain. Does this also oscillate under the here-used experimental conditions?

-Single cell traces of the Phsp-YFP are now included (see Fig. EV1D-F). As expected, no significant pulsing is seen for this reporter as its expression is independent of pulsing Spo0A activity (see lines 123-125).

10. L280, bacteria do not choose, they can't think.

-The terms "cell-fate choice" and "cell-fate decisions" are widely used in the field. We've replaced "choose" by "decide" (line 301).

11. Can the authors speculate why there is so much heterogeneity in growth rates in these otherwise isogenic cells grown under more or less identical conditions?

-The source of growth rate heterogeneity can't be deduced from our data as this heterogeneity originates outside of the sporulation network. However, the fact that growth rates of sister cells are only weakly correlated suggests some sort of intrinsic noise as a driver of this heterogeneity. Perhaps stochastic fluctuations in the rate-limiting enzymes in cellular metabolism could be a possible explanation. Following the reviewer's suggestion, we've included a short speculation on this issue in the discussion section (see lines 557-564).

12. Is growth and sporulation heterogeneity observed under more uniform conditions such as within microfluidic devices that can keep the nutrient, oxygen and waste flow uniform?

-As indicated by low correlations between sister-cells which are right next to one another, the heterogeneity is unlikely to originate from spatial non-uniformity on our microplates. Investigating the extent of the growth rate fluctuations on microplates and in microfluidic devices would be a subject of the future studies.

13. L454, This was a very complicated way to show that KinA does not require an external signal to activate sporulation. Here it should be noted that similar conclusions have been made previously by the authors (e.g. Devi et al, 2015 JBact, Eswaramoorthy Jbact).

-We thank the reviewer for raising this concern since this is an important point. Our aim in the Results section "KinA activity does not depend on the growth rate" was to go beyond the results of Devi et al, JBact 2015 and Eswaramoorthy et al., JBact 2011. Those studies do indeed establish that

no external signal is *essential* for KinA to activate sporulation. However they do not rule out the possibility that some non-essential signals are playing a role in sporulation by modulating KinA activity in a growth dependent manner. In contrast our results conclusively show that the activity of KinA (or other phosphorelay proteins) is not modulated in growth-dependent manner. If KinA activity was being significantly modulated, this would be reflected in IPTG dependent KinA thresholds as explained in Fig. 5. Since the same threshold of KinA required for sporulation independent of growth rates in our conditions, no signal affects KinA activity as cells starve. To avoid confusion we have clarified the motivation for these experiments at the start of this section (see lines 416-425).

14. L509, this statement is incorrect. It has not been shown that bet-hedging or memory controls sporulation. Rather, it has been suggested that heterogeneity in spore formation might provide a bet-hedging strategy (see e.g. de Jong et al. 2011 Bioessays). The here-discovered growth-rate heterogeneity driving the sporulation decision might still be part of an evolutionary bet-hedging strategy. Also, the authors have not tested whether the slow growth phenotype can be inherited (see my point 4).

-We have rephrased this passage to avoid confusion (lines 552-557). We can conclude that contrary to an earlier claim, the phosphorelay network is not the “noise generator” responsible for bet-hedging. Heterogeneity is generated on the level of growth rate control, upstream of the sporulation network.

Reviewer #2:

This report by Naruta et al is the latest in a fascinating series of studies from the same group that have shed much light on the complex regulation of sporulation in B. subtilis. The authors have used a combination of modeling and wet experiments to support an enticingly simple model for the crucial signal for the onset of spore formation. It is suggested that starvation does not result in a chemical signal that activates the phosphorelay, but that the slowdown in cell growth that results from starvation automatically boosts the concentration of key regulatory proteins, causing the accumulation of Spo0A-P, the master activator of spore transcription. The authors propose that this is the "primary signal for sporulation" and that all else is just a matter of "checkpoints and fine tuning".

The data in this report are ingenious and convincing and this paper should have a major impact. But there are a few points that are either confusing to me or perhaps merit discussion. I would argue that although the authors have made strong case, it has been unnecessarily overstated.

1. The Herman lab has recently published a paper in PLOS One that uses the artificial induction of KinA to show that the initiation of spore formation cannot proceed when the culture is growing exponentially, but does take place when growth slows. This is completely consistent with the model advanced from the present study. However, when conditioned medium is added, containing a so-called factor X, cells in an exponentially growing culture can proceed at least through engulfment. This would seem to be contrary to the expectations from the present study. I realize that the Herman paper does not examine the rate of increase of cell volume as sporulation initiates, but rather the ensemble rate of change of OD. But I think this apparent contradiction deserves comment.

-We thank the reviewer for bringing this to our attention. Indeed the results of Ababneh et al., are very interesting and deserve comment. We find no contradiction between the results of Ababneh et al. and our own results. As pointed out in their study, FacX appears to be an essential factor that promotes sporulation by a mechanism that is independent of the phosphorelay. As a result, in its absence KinA induction in exponential phase cells is unable to trigger sporulation. In its presence, exponentially growing cells still require KinA overexpression to sporulate i.e. the higher growth rates need to be compensated for by overproducing KinA. Thus the growth threshold model is consistent with the results of Ababneh et al. – we just need to account for the fact that FacX might be independent essential requirement for sporulation. We have added a paragraph to our discussion section to discuss these implications of their findings for our growth model (see lines 601-615).

2. It is well known that certain media favor sporulation and others do not. For example, the addition of glucose sharply reduces sporulation. And other media lacking glucose apparently do not allow much sporulation immediately after cells enter stationary phase. One such example is MSgg, commonly used to study biofilm formation. Doesn't this suggest that there are important nutritional signals that are independent of growth rate? Again, I think comment is needed.

-We've discussed the effects of media and possible "permissive" signals for sporulation in the expanded discussion section (see lines 592-600).

3. I find the following to be confusing. On lines 324 and following, the authors describe experiments with their *iTrans-0F* strain. It is stated on line 330-333 that when IPTG is omitted the gene dosage ratio of *kinA/0F* is "1" and that as a result, pulses of *0A~P* do not occur as a result of DNA replication. This is understandable. But the reason for the pulses is that when the ratio is 0.5, as in the wild-type strain before the completion of replication, the excess of *0F* inhibits the phosphorylation of *KinA*. So, in the *iTrans-0F* strain growing without IPTG, *KinA* should not be inhibited. Consequently, *KinA~P* should be formed continuously at a high rate and sporulation should take place as growth slows. Why is it stated that the strain will not sporulate without IPTG?

-We apologize for the confusion. The observed behavior of *iTrans-0F* strain was explained in detail in Narula et al, Cell 2015 and therefore was omitted here. In the absence of IPTG, *iTrans-0F* strain behaves the same as *Trans-0F* strain in which *0F* was translocated to a position near *KinA*. As predicted by our modeling work and demonstrated by the follow-up experiments, in that strain lack of pulsing is associated with the lack of "trigger" that make system overshoot. Using the mechanical analogy – for the pendulum to swing to the right it first must be tilted to the left. In wild-type cells during DNA replication *0A~P* is below its steady state corresponding to 1:1 *KinA:0F* ratio. However, upon completion of DNA replication as *KinA:0F* ratio returns to 1:1 the *0A~P* swings past that steady state due to delayed negative feedback in the network. We've slightly expanded this explanation and referred the reader to Cell paper for details (see lines 350-360).

4. On lines 525 and following it is stated that knocking out the genes listed on line 518 does not significantly affect sporulation. This is to counter the potential argument that the growth rate model is an over-simplification. However, although eliminating *KipA* does not inhibit sporulation, the elimination of *KipA*, a *KipI* antagonist, does reduce spore formation. This pathway is thought to respond to the availability of nitrogen. *Sda* inhibits sporulation when replication initiation is inhibited; there is a DNA replication checkpoint. *PhrA* is but one anti-*Rap* peptide. When the permease for these peptides is eliminated, sporulation is impaired. So quorum sensing, aside from Herman's *FacX*, is an important input. True, a *yheH* knockout is not reduced in sporulation, but *yheI/yheH* overexpression does lower spore frequency. Could this hint at a role for this ABC transporter in sensing an unknown environmental signal? I think that the growth rate model is quite pretty but we must be careful not to be seduced into embracing an oversimplification. A decision as drastic for the cell as embarking on spore formation is likely to be hemmed in by all sorts of positive and negative signals in addition to decreasing growth rate.

-Following the reviewers comments we have modified and reorganized our discussion section to place the role of growth rate dependent sporulation control in the proper context of other regulators and environmental/nutrient signals (see lines 581-615). We first point out that growth rate likely functions along with other signals/regulators in controlling cell-fate in different conditions. We also suggest that the growth rate model may be a useful way of thinking about how multiple environmental signals may be effectively integrated to control the sporulation decision. Specifically these regulators may either change the growth rate threshold for sporulation in a nutrient/cell density dependent manner or alternatively act as all-or-none checkpoints that prevent even slow growing cells from activating *0A*. We also indicate how growth rate may be a misleading signal when cells are under stress (low temperature, antibiotics, ethanol etc. can decrease growth) and that the other regulators of sporulation may play a critical role by preventing cells from sporulating under such conditions.

Corresponding Author Name: Oleg A. Igoshin

Manuscript Number: MSB-15-6691